# The Effect of Encapsulated Apigenin Nanoparticles on HePG-2 Cells through Regulation of P53

**DOI:** 10.3390/pharmaceutics14061160

**Published:** 2022-05-29

**Authors:** Mayada Mohamed Mabrouk Zayed, Heba A. Sahyon, Nemany A. N. Hanafy, Maged A. El-Kemary

**Affiliations:** 1Chemistry Department, Faculty of Science, Kafrelsheikh University, Kafrelsheikh 33516, Egypt; mayadazayed556@gmail.com (M.M.M.Z.); heba_sahuon@sci.kfs.edu.eg (H.A.S.); 2Nanomedicine Group, Institute of Nanoscience and Nanotechnology, Kafrelsheikh University, Kafrelsheikh 33516, Egypt; elkemary@nano.kfs.edu.eg

**Keywords:** hepatocellular carcinoma, apigenin, capsase, apoptosis

## Abstract

Apigenin (Ap) is one of the most important natural flavonoids that has potent anticancer activity. This study was designed, for the first time, to load Ap into chitosan to improve its hydrophobicity and then it was coated with albumin-folic acid to increase its stability and bioavailability and to target cancer cells. The newly developed encapsulated Ap (Ap-CH-BSA-FANPs) was characterized and tested in vitro. The zeta potential of −17.0 mV was within the recommended range (−30 mV to +30 mV), indicating that encapsulated apigenin would not quickly settle and would be suspended. The in vitro results proved the great anticancer activity of the encapsulated apigenin on HePG-2 cells compared to pure Ap. The treated HePG-2 cells with Ap-CH-BSA-FANPs demonstrated the induction of apoptosis by increasing p53 gene expression, arresting the cell cycle, increasing caspase-9 levels, and decreasing both the MMP9 gene and *Bcl-2* protein expression levels. Moreover, the higher antioxidant activity of the encapsulated apigenin treatment was evident through increasing SOD levels and decreasing the CAT concentration. In conclusion, the Ap-CH-BSA-FANPs were easy to produce with low coast, continued drug release, good loading capacity, high solubility in physiological pH, and were more stable than the formerly Ap-loaded liposomes or PLGA. Moreover, Ap-CH-BSA-FANPs may be a promising chemotherapeutic agent in the treatment of HCC.

## 1. Introduction

Hepatocellular carcinoma (HCC) is the sixth most frequent cancer and the third leading cause of cancer-related deaths in the world. The majority of HCC cases are primarily caused by hepatitis B or C infection [1]. However, various other reasons have been identified, including alcohol consumption, oxidative stress, toxic materials, and food carcinogens such as aflatoxins and nitrosamines [2]. The most common methods for treating HCC are radiation and chemotherapy. Doxorubicin (DOX), 5-fluorouracil, and cisplatin (Cis.) are the most common chemotherapeutic drugs that have been used to treat HCC [3,4]. Heart and kidney failure are serious adverse effects of chemotherapeutic drugs that might make it difficult to treat HCC successfully [5,6]. Therefore, researchers have tried to invent new strategies for HCC treatment with natural flavonoids. Indeed, the combinations of polyphenols with either DOX or Cis. have the ability to reduce their toxicity and increase their efficiency [7,8,9].

Natural flavonoids have been widely used in cancer prevention as well as in treatment. Apigenin (Ap) is one of the most important flavonoids that has potent anticancer, anti-inflammatory, and antioxidant activities [10,11,12,13]. Ap is found in large quantities in oranges, onions, and parsley. Ap or 4′, 5,7-trihydroxyflavone has a chemical formula of C15H10O5 and a molecular mass of 270.24 g mol. The term “apigenin” is derived from the Apium genus of the Apiaceae family. 

Several studies have reported that the anticancer activity of Ap is due to its ability to induce apoptosis and cell cycle arrest [14,15,16,17]. Unfortunately, Ap has limited water solubility and bioavailability [18], owing to its low absorption and metabolism. For this reason, various efforts have been made to develop nanocarriers to improve its bioavailability and its therapeutic benefits. Recently, for the delivery of apigenin, liposomes, and PLGA nanoparticles have been created [19,20]. The nano-formulation of Ap has improved its therapeutic efficacy, anti-tumor activity, and cytotoxicity against tumor cells. Mahmoudi and coworkers (2019) confirmed that the nanostructured lipid carriers encapsulating Ap had increased cytotoxicity against lung cancer cells and enhanced docetaxel’s effectiveness when combined with it [21]. Another study revealed that the Ap-loaded PLGA had increased the diffusion of Ap into skin cancer cells, leading to increased apoptosis [22]. Moreover, loading Ap on galactose-PLGA nanoparticles had an enhanced apoptotic and cytotoxic effect against HePG-2 cells [20]. However, liposomes’ high manufacturing cost, limited lifetime, reduced stability due to oxidation, and even lower solubility may limit their usage in large-scale cancer therapy [23]. Alternatively, encapsulation with PLGA could cause the encapsulated drug to aggregate, leading to limited drug loading [24,25]. Furthermore, PLGA has been shown to be a passive targeted vehicle incapable of delivering appropriate drug concentrations to cancer cells [26]. 

Chitosan nanoparticles (ChNPs) have the ability to carry chemotherapeutic medicines into cells, increasing their activity and releasing the encapsulated compounds over time [27,28]. When polyphenols are loaded onto ChNPs, the –OH groups of the polyphenols react with the ChNPs to generate new bioactive nano-compounds [29,30,31]. Many polyphenolic compounds are loaded onto ChNPs as a stable, low coast, biodegradable drug delivery vehicle, but its low solubility at physiological pH is still a problem in drug distribution inside the body [32]. According to our knowledge, only one study was previously published that described the loading of Ap into chitosan-stearate nanogel and was used to increase its apoptotic activity, but no molecular pathways were detected [33].

Bovine serum albumin (BSA) can be used to improve the solubility of hydrophobic drugs [34]. Coating ChNPs with BSA can improve their stability as well as their water solubility [35]. It was recently demonstrated that loading resveratrol onto ChNPs coated with BSA increased the antioxidant activity of that polyphenol, reduced its degradation, and increased its release within the cell [36]. Moreover, albumin is able to target tumor cells by interacting with albumin-binding proteins such as secreted protein acidic and rich in cysteine (SPARC) and glycoprotein 60 (gp60) [37]. 

In this study, we aimed to load Ap onto ChNPs and then coat them with albumin-folic acid to increase its bioavailability, solubility, and stability and may bind it to the target tumor cells by both albumin-binding protein and folate receptor (FR)-mediated endocytosis. FR was highly expressed in human malignancies with low expression in normal tissues [38,39,40]; hence the resultant encapsulated apigenin (Ap-CH-BSA-FANPs) may have targeted the cancer cells more than normal once. The developed encapsulated apigenin (Ap-CH-BSA-FANPs) are inexpensive and are expected to be water soluble and highly stable with great influence to release the Ap continuously. Moreover, the resultant encapsulated apigenin was tested to see if it could be used together with either doxorubicin or cisplatin or as a substitute for these drugs. The Ap-CH-BSA-FANPs were characterized and subjected to the in vitro assays. Furthermore, the apoptotic/necrotic pathway of the encapsulated apigenin on the HePG-2 cells was assessed, and the change in the p53, MMP9, caspase 9, and *Bcl-2* gene expressions will be detected.

## 2. Materials and Methods

### 2.1. Chemicals and Reagents

Chitosan, dimethyl sulfoxide (DMSO), bovine serum albumin (BSA), folic acid (FA), Doxorubicin (DOX), cisplatin (Cis.), N-Ethyl-N′ -(3-dimethylamino propyl) carbodiimide hydrochloride (EDAC), N-Hydroxysuccinimide (NHS), and sodium hydroxide pellets were purchased from Sigma Aldrich, St. Louis, MO, USA; ethanol was obtained from Baker Analyzed, Fisher Scientific, Landsmeer, The Netherlands. Apigenin (99.06% purity, CAS Number 520-36-5) was obtained from Chengdu Herbpurify Co., Ltd. (Wuhan, China). Hepatocellular carcinoma (HePG-2) and epithelioid carcinoma (Hela) cell lines were obtained from ATCC via Holding Company for Biological Products and Vaccines (VACSERA), Cairo, Egypt. Fetal Bovine serum (GIBCO, Carlsbad, UK), Annexin V-FITC Apoptosis Kit Plus BioVision. Qiagen RNA extraction/BioRad SYBR green PCR MMX. A catalase activity assay colorimetric kit (ab83464) and propidium iodide flow cytometry kit (ab139418) were purchased from Abcam. Superoxide dismutase (SOD) activity assay: Enzyme-linked Immunosorbent Assay Kit (Cloud-Clone Corp). All other reagents used were of analytical grade.

### 2.2. Fabrication of Albumin-Folic Acid

The carbodiimide reaction was used to activate the carboxylic group of folic acid [41]. Briefly, a stock solution of activated FA was prepared by dissolving 10 mg of FA in 2.5 mL of DMSO under continuous stirring until completely soluble. NHS (10 mg) and EDAC (30 mg) were added to the FA solution to complete the solution to 5 mL with DMSO and stirred for 30 min. A weight of 1 mg of BSA (1 mg/mL) was dissolved in 1 mL of distilled water. Finally, 1 mL of activated FA solution was slowly added to the BSA solution and stirred for 30 min. Alternatively, NaOH (1 N) can be used to dissolve and activate FA as well [41].

### 2.3. Fabrication of Apigenin NPs 

A weight of 1 mg of chitosan was first dissolved in 4 mL of 1% glacial acetic acid to reach a concentration of 0.25 mg/mL. After that, 3 mg of Ap was dissolved in 10 mL of 1 N NaOH and dropped slowly into 10 mg/40 mL of chitosan under continuous stirring for 30 min. Additionally, 20 mL of BSA (1 mg/mL) conjugated to FA was added to the mixture and stirring was completed for 30 min. The formed nanoparticles were dialyzed against distilled water for 24 h before being stored at −20 °C for lyophilization. The resultant nanoparticles were freeze-dried on Martin Christ lyophilizer at −58 °C and 0.3 mbar for 72 h.

#### Characterization 

The characterization of encapsulated apigenin was assessed by Transmission Electron Microscopy (TEM) (JEOL 2100, Ltd., Tokyo, Japan) operating at an accelerating voltage of 200 kV, Scanning Electron Microscope (SEM) (JEOL, JSM–IT 100), and mages were processed by using the software SEM/JSM 5000 and UV–vis spectroscopy (Jasco V-770 UV Visible Absorbance Spectrophotometer. Zeta Potential was determined by Brookhaven using a Zeta Nano Sizer and Zeta Potential. Fourier Transform Infrared Spectroscopy (FTIR) JASCO, Tokyo, Japan, model no. AUP1200343) was used to detect the surface molecular structures in the range of 500–4000 cm^−1^ by using the KBr pellet method. X-ray diffraction was detected by Shimadzu (XRD 6100) diffractometer with CuKa1 radiation (k = 1.54056 A°). 

### 2.4. Determination of Encapsulation Efficiency

After the encapsulation process, the supernatant was used to detect the loading capacity of the encapsulated Ap by using a UV visible spectrophotometer using the Ap standard curve method. Therefore, the pure Ap dissolved by DMSO was subjected to spectrophotometric analysis at 270 nm. The concentration of Ap was determined by following the known serial concentration of Ap at (10 μg to 60 μg) (R2 = 0.9967) [42], and the % of encapsulation efficiency was then calculated using the following equations; 

Encapsulation efficiency (%) = Conc. of total Ap-Conc. of free Ap (supernatant)\Conc. of total Ap × 100.

### 2.5. In Vitro Drug Release Study 

The in vitro release study of the Ap from Ap-CH-BSA-FANPs and pure Ap was performed in the presence of a chemical initiator such as DMSO or in physiological media according to the method of Hanafy et al. 2021 [43,44]. 

The pure Ap and encapsulated Ap were released in PBS (pH 7.4, representing the physiological environment) and PBS (pH 6.5, representing the cancer environment) containing 1% Tween 80 for more than 50 h. Briefly, 16 mg of Ap pure and (Ap-CH-BSA-FANPs) suspensions were transferred into dialysis bags (molecular weight 12,000–14,000 Da) and then immersed in 40 mL release medium at 37 °C with continuous stirring at 100 rpm. Following that, 1 mL was aspirated from each baker to determine the quantity of apigenin released at 0, 1, 2, 4, 6, 12, 24, and 48 h. At the same time, the baker’s solution was replaced after every aspiration with the same volume of freshly prepared release medium. The absorbance of Ap was measured using a UV-Vis spectrophotometer at a wavelength of 270 nm.

The % of cumulative drug release was measured using the constructed standard calibration curve. The mean release of three measurements was used to evaluate the drug release for each point.

Concentration of drug (µg/mL) = (slope × absorbance) ± intercept

Amount of drug = Concentration × Dissolution bath volume × dilution factor released mg/mL 1000

Cumulative percentage = Volume of sample withdrawn (mL) × P (t − 1) + Pt release (%) Bath volume

(v) where Pt = Percentage release at time t Where P (t − 1) = Percentage release previous to ‘t’

### 2.6. Cellular Uptake and Targeting Capacity 

HePG2 cells (10^6^) were seeded upon the surface of a sterilized coverslip and laid on the bottom of 6 multi-well plates. After 24 h from their growth, 50 μg/mL (Rhodamine-Ap-CH-BSA-NPs) with or without FA was added to each well and incubated for 24 h. in a humidified atmosphere of 37 °C, 5% CO_2_. HePG2 cell lines were washed with phosphate buffer saline and then fixed with 4% paraformaldehyde. Cells were further washed by PBS,7.2. Cellular uptake was analyzed after 24 h by red (TRITC) channels of fluorescence microscopy, and then images were captured by a digital camera. 

Targeting capacity was demonstrated by the intensity of accumulated NPs in the perinuclear region of the cytoplasm.

### 2.7. MTT Assay

The MTT assay was used to assess the inhibitory effects of certain compounds on cancer cell growth. The cytotoxic effects of Free capsules, Ap, Ap-CH-BSA-FANPs, Ap-CH-BSA-FANPs + DOX, and Ap-CH-BSA-FANPs + Cis. were performed against HePG-2 and Hela cell lines. The colorimetric test is based on using the mitochondrial succinate dehydrogenase to convert yellow tetrazolium bromide to a purple formazan derivative in cancer cell lines. Cancer cells were replicated in RPMI-1640 media containing 10% fetal bovine serum at 37 °C in a 5% CO_2_ incubator. Different cancer cells were seeded with/without Free capsules, Ap or Ap-CH-BSA-FANPs, at a density of 1.0 × 10^4^ cells/well in a 96-well plate, then incubated at 37 °C for 48 h in a 5% CO_2_ incubator. After 48 h of treatment, 20 µL of 5 mg/mL MTT solution was added and re-incubated for 4 h. To dissolve the purple formazan produced, 100 µL DMSO was added to each well; then, the color intensities were measured with a plate reader (EXL 800, USA) at 570 nm. The IC_50_s were calculated and compared to the IC_50_s of DOX. and cisplatin. IC_50_s were calculated from the following equation;

Relative cell viability percentages = [(A_570_ of treated samples/A_570_ of untreated samples) × 100].

### 2.8. Annexin/PI Assay

The apoptotic effect of Ap or Ap-CH-BSA-FANPs on HePG-2 cells was evaluated using a flow cytometer technique and the Annexin V-FITC apoptosis detection kit. HePG-2 cells (1 × 10^5^) were planted in a culture flask and incubated overnight in a 5% CO_2_ and 95% humidity environment. The cells were treated with the IC_50s_ values of either Ap or Ap-CH-BSA-FANPs and incubated for 48 h. The treated cells were then collected and suspended in 1% ice-cold PBS before being tagged with 5μL FITC-conjugated Annexin V/PI, in a dark environment, under steady stirring. A flow cytometer was used to examine the treated/untreated labeled cells.

### 2.9. Cell Cycle Examination 

Flow cytometric analysis was used to determine the cell distribution ratios in the G1, S, and G2/M phases of the cell cycle. HePG-2 cells (3 × 10^5^ cells/800 μL/well) were seeded in 6-well plates and incubated overnight. The cells were then treated with the IC_50_ doses of Ap or Ap-CH-BSA-FANPs and incubated at 37 °C for 48 h. Then, the cells were carefully washed with PBS and centrifuged. The cell’s pellets were then stained with the Flow Cytometry kit (cat. no. ab139418; Abcam) according to the kit’s protocol. The data were analyzed using FACSCalibur, BD Bioscience, San Jose, CA, USA. The results were compared to the untreated HePG-2 cells. 

### 2.10. Real-Time PCR

Total RNA was extracted from the untreated/treated HePG-2 cells with the IC50 of Ap or Ap-CH-BSA-FANPs using Qiagen RNA extraction/BioRad SYBR green PCR MMX according to the manufacturer’s instructions. The Rotorgene RT-PCR system was used to assess the concentration of isolated RNA. The p53 primers used were (F 5′-ATGTTTTGCCAACTGGCCAAG-3′, R 5′-TGAGCAGCGCTCATGGTG-3′, Casp9 primers were (F 5′-CATTTCATGGTGGAGGTGAAG-3′, R 5′-GGGAACTGCAGGTGGCTG-3′) and the MMP9 primers were (F 5′-CCA CGG TGC GGG GTC CCA GAC-3′, R 5′-GGA GAC GCC CAT TTC GAC GA-3′. The β-actin primers were (F 5′-GTGACATCCACACCCAGAGG-3′, R 5′-ACAGGATGTCAAAACTGCCC-3′).

### 2.11. Immunohistochemical Analysis of Bcl-2 Protein

HePG-2 cells were treated with Ap or Ap-CH-BSA-FANPs for 48 h. Cells were then collected and centrifuged for 10 min at 1700 rpm in cooling centrifuges. The supernatant was properly aspirated, and the cell pellets were washed three times with sterile PBS. A volume of 50 µL of treated/untreated cells was properly aspirated into clean, positively charged glass slides before being stored in the refrigerator overnight. All of the slides were fixed with methanol for 30 min. After fixation, the slides were incubated for one hour at 24 °C with a primary anti- *Bcl-2* antibody (R & D Systems Inc., MAB827, Minneapolis, MN, USA). The slides were then immersed three times in PBS, incubated for 30 min at room temperature with anti-mouse IgG secondary antibody (EnVision + System HRP; Dako, Agilent, Santa Clara, CA, USA) visualized with diaminobenzidine commercial kits (Liquid DAB + Substrate Chromogen System; Dako), and counterstained with Mayer’s hematoxylin. Finally, a light microscope was used to examine the slides. The number of positive cells in a total of 1000 counted cells in at least 10 high power fields was used to calculate the *Bcl-2* protein.

### 2.12. Antioxidant Assay

The activities of catalase (CAT) and superoxide dismutase (SOD) enzymes of untreated/treated HePG-2 cells were assessed by using the Abcam colorimetric kit (ab83464) and the Cloud-Clone Corp (Enzyme-linked Immunosorbent Assay) Kit. The tested enzymes were determined according to the manufacturer’s protocol.

### 2.13. Bio-Statistical Analysis

The results were expressed as mean ± standard error of mean (SEM). Data were analyzed by SPSS 18 program using one-way analysis of variance (ANOVA), followed by Duncan’s test for comparison between different treatment groups. Statistical significance was set at *p* ≤ 0.05.

## 3. Results 

### 3.1. Size, Shape and Characteristics of the Encapsulated Apigenin 

Chitosan is a promising polymer for encapsulation polyphenolic compounds because it includes several amino groups (–NH_2_) that can be protonated (–NH_3_^+^) at acidic pH during its dissolution. Chitosan’s cationic charge permits it to increase the loading capacity of polyphenolic compounds. Apigenin was dissolved in 1 N NaOH and then mixed with chitosan for 15 min. [45]. Apigenin has been deprotonated in an alkaline solution, resulting in a highly negative charged molecule. As a result, a network hydrogel was formed between the chitosan and Ap. The hydrogel core was then coated with folic acid conjugated BSA to help with targeted delivery [43], and its solubility in physiological pH was increased. Because BSA has a high concentration of charged amino acids, especially lysine, active folic acid can quickly interact with BSA’s amino groups [46].

TEM images showed uniform and well dispersed spherical and semi-spherical encapsulated apigenin with a size of almost 30–40 nm Figure 1A SEM images showed micro/nanopores were integrated inside moieties of the assembly Figure 1B,C. 

The main characteristic peaks of folic acid adsorption were mostly estimated at 276 and 357 nm [47]. However, the current study showed that the three peaks of pure folic acid were measured at 256 nm, 281 nm, and 367 nm because of its dissolution at alkaline ionic strength. Furthermore, its carboxyl groups have been deprotonated. At the same time, these peaks were re-modulated to their adsorption at 269 nm and 363 nm after their interaction with BSA and chitosan (Figure 2(A1)). These data strongly confirmed the presence of folic acid conjugated nanoparticles. 

The mean characteristic peaks for the adsorption of apigenin were measured at 268 nm and 337 nm [48], while a broad peak was estimated at 356 nm, which was attributed to apigenin and folic acid cross-linking (Figure 1 and Figure 2(A2,A3)). As a comparison, there was a shift in the adsorption peak of pure apigenin from 268 nm to 279 nm after its encapsulation (Figure 2(A3)). The percent of encapsulating apigenin was measured by using the standard curve of pure apigenin (Figure 2(D1,D2)), and the encapsulation efficiency of apigenin was found to be around 69% (Figure 2C).

The prepared nanoparticles were characterized on the basis of particle size, size distribution, and zeta potential. The mean particle size of the free capsule was 100 nm with a PDI of 0.1 ± 0.004, while encapsulation of Ap exhibited a mean particle size of 189 nm and a PDI of 0.3 ± 0.2. Low PDI values indicate that the formulations exhibit even-sized particles (Figure 3A). The mean zeta potential values of the free capsule and Ap-CH-BSA-FANPs, as determined by the DLS technique, were found to be −26 and −17 mV, respectively. The zeta potential of −17.0 mV was within the recommended range (−30 mV to +30 mV), indicating that encapsulated apigenin (Ap-CH-BSA-FANPs) would not quickly settle and would be suspended and distributed in the physiological media and could be injected intravenously [49]. The negative zeta potential also favors reticuloendothelial system (RES) uptake by the liver and spleen [17] (Figure 3B).

FTIR is used to study the modification of chemical bands before and after conjugation. The FTIR spectrum of chitosan revealed that the O-H and N-H were assigned to the 3514 cm^−1^ band and 3034 cm^−1^ band, respectively, while the 1675 cm^−1^ band was attributed to amido groups. The bands located at 1087 cm^−1^ to 883 cm^−1^ were ascribed to the β-1,4 glycoside bond (Figure 4A) [50,51].

A major band was observed at 3365 cm^−1^ for pure BSA (amide A, related to N-H stretching), 2976 cm^−1^ (amide B, N-H stretching of NH_3_ + free ion), 1663 cm^−1^ (amide I, C = O stretching), and 1514 cm^−1^ (amide II, related to C-N stretching and N-H bending vibrations). The most intense bands are associated with the secondary structure and conformation of proteins [52].

The FTIR of the FA spectrum showed bands at 3525 and 3163 cm^−1^ due to O-H and N-H stretching vibrations, respectively [53]. The FA spectrum showed a peak at 1702 cm^−1^, revealing a carboxyl group.

The FTIR of the BSA-FA spectrum (Figure 5B) showed a band at 1702 cm^−1^ that was attributed to the carboxylic group of FA. Additionally, the characteristic bands of p aminobenzoic acid were observed at 1404 cm^−1^ to 1100 cm^−1^.

The FTIR of the free capsules spectrum showed bands at 3432 cm^−1^ that were attributed to O-H and N-H. The observed band at 2910 cm^−1^ was assigned to the stretching vibration of the folic acidpterin ring. There was a presence band at 1650 cm^−1^ that was associated with the stretching vibration of the carboxyl-amid bond attachment of BSA-CHI.

There was also a band at 1541 cm^−1^ that indicated the presence of BSA amid II. Another band, observed and located at 1032 cm^−1^, was assigned to the starching vibration of the C-H backbone of chitosan. 

However, the bands observed between 1408–1032 cm^−1^ were assigned to the presence of an aromatic ring stretch of the pyridine and p-amino benzoic acid moieties [54]. The spectra of pure Ap showed that there was a broad band at 3299 cm^−1^ originating from the valence vibration of (O-H) groups in the structure of pure Ap, which was likely to be related to the formation of intramolecular hydrogen bands with the C=O group of the ring. Some obvious intensive bands at 1650, 1608, and 1514 cm^−1^ could prove the existence of C=O groups in the structure of raw Ap. The other intensive band at 845 cm^−1^ was generated as a result of the C-O groups and deformation (C-OH) variation from the structure of raw Ap [49].

The FTIR of the Ap-CH-BSA-FANPs spectrum showed bands at 3423 and 3057 cm^−1^ that were associated with O-H and N-H starching vibration. The band located at 2910 cm^−1^ was assigned to the pterin ring structure of FA (Figure 4B).

X-ray diffraction is an advanced technique that can identify the crystallinity of material at the atomic scale. The diffraction of pure apigenin has a natural crystallinity showing multiple peaks at angles 2θ = 9°, 10°, 13.5°, 14.5°, 15°, 17.5°, 20°, 22°, 23°, 24.7°, 26.8°, 31.5°, 37°, and 23°. The peaks of folic acid were patterned at angles of 2θ = 10°, 12.6°, 16°, 21°, 26°, 28.9°, and 37°. The spectrum of free capsules with no apigenin shows peaks at 2θ = 8°, 18.6°, 26.6°, 29°, and 37°. The XRD pattern of encapsulated apigenin contains new peaks at 2θ = 8.9°, 10.9°, 18.6°, 22°, 29°, and 37°. Additionally, one shared peak at 2θ = 37° was represented by FA, free capsules, and encapsulated apigenin (Ap-CH-BSA-FANPs).

### 3.2. In Vitro Drug Release

An in vitro study revealed that encapsulated apigenin exhibited continuous release during incubation at time intervals. This strategy depends mainly on the amount of apigenin entrapped in chitosan moieties and the interaction of the amino groups of chitosan and the hydroxyl groups of apigenin after their de-protonation. Due to its hydrophobic nature, apigenin pure showed good release in the presence of a chemical imitator such as DMSO. The drug release was 40% after 24 h incubation. At the same time, less amount of release was obtained by Ap-CH-BSA-FANPs (20%) in the presence of the same initiator. This is due to the chemical interaction between entrapped Ap inside moieties of chitosan and BSA. Contritely, slow release was observed by free apigenin at physiological pH 7.4 and tumor microenvironment pH 6.5. This is due to its solubility in an aqueous solution. Meanwhile, the release of Ap from nano-formulation (Ap-CH-BSA-FANPs) was quite higher in buffer of pH 6.5 compared to PBS (pH 7.4). 

The cumulative drug release of encapsulated Ap. was around 85% at pH 6.5, while it was around 60% (at pH 7.4) at the end of 50 h incubation. The hydroxyl groups of apigenin interacted with the amino group of the chitosan forming stable structure. Such interaction can be ionized at acidic pH, thereby releasing the drug effectively at acidic pH suitable to target cancer cells.

In the current study, apigenin was released continuously throughout the time of incubation, providing control release compared to free apigenin (Figure 6). In the meantime, our results revealed the possible release of apigenin encapsulated inside moieties of chitosan after its cellular uptake. 

### 3.3. Demonstration of the Targeting Capacity of the Nanoparticles

The targeting capacity of Ap-CH-BSA-FANPs internalized inside HePG2 cells was measured by fluorescence microscopy. R6G labeled Ap-CH-BSA-FANPs (Figure 7A,B) were successfully localized inside the cytoplasm, as demonstrated by the intensity of fluorescence emission of red color located in the perinuclear region compared to Ap-CH-BSA-NPs without FA (Figure 7C).

Fluorescence images clearly demonstrate that Ap-CH-BSA-NPs conjugated to FA were readily accumulated in cellular compartments leading to increase drug capacity and efficiency. 

### 3.4. In Vitro Cytotoxic Activity

The anticancer effect of free capsules, Ap, Ap-CH-BSA-FANPs, DOX, and Cis. as well as Ap-CH-BSA-FANPs + DOX and Ap-CH-BSA-FANPs + Cis, were tested, in triplicated manner, against HePG-2 and Hela human cell lines. The anticancer activities were evaluated by MTT method to estimate the IC_50s_. The synergistic or antagonistic effect of the Ap-CH-BSA-FANPs with both DOX and Cis, standard anticancer drugs, were also assessed by the same method. All tested compounds, DOX and Cis. were diluted into different concentrations (100–1.56 μg/mL), and every concentration was mixed in a well containing 1.0 × 10^4^ cells, then incubated for 48 h. Finally, the IC_50_ values were evaluated and matched with the IC_50_ of the DOX and Cis.

Figure 8 showed the viability percentage of both HePG-2 and Hela cells was decreased gradually with increasing the Ap-CH-BSA-FANPs concentration, which indicates the anti-proliferative effect of that compound in a concentration-dependent manner. Our data revealed a marked and significant decrease in the IC50s of the encapsulated apigenin (Ap-CH-BSA-FANPs) on both HePG-2 and Hela cells compared to free apigenin (*p ≤* 0.0001, Table 1). These data confirmed the increased anticancer activity of Ap-CH-BSA-FANPs. However, high IC_50_s values were observed on both HePG-2 and Hela cells treated with free capsule nanoparticles, which indicated its non-antiproliferative effect (Table 1).

On the other hand, the IC_50_s of the Ap-CH-BSA-FANPs were significantly increased on both HePG-2 and Hela cells compared to both DOX and Cis. However, a non-significant change was observed in the HePG-2 cells treated with the encapsulated apigenin compared to Cis (*p* = 0.152), which confirmed the potent anticancer activity of Ap-CH-BSA-FANPs. Similarly, a non-significant change was detected in the HePG-2 cells treated with the Ap-CH-BSA-FANPs + DOX compared to DOX alone (*p* = 0.152), which confirms a synergy between encapsulated apigenin and DOX (Table 1). Recently, Mahmoudi et al. (2019) reported that the combination of nano-lipid carriers carrying apigenin and the anticancer drug docetaxel resulted in a reduced A549 cancer cell proliferation [21].

Our data established the great anticancer activity of the Ap-CH-BSA-FANPs on HePG-2 cells, which can be combined with DOX to reduce its toxicity with good efficiency. Moreover, the IC_50_ of Ap-CH-BSA-FANPs + DOX on HePG-2 cells was nearly the same as that of the Cis., which indicates the potency of that combination as an anticancer agent.

### 3.5. Ap-CH-BSA-FANPs Treatment Promotes HePG-2 Cell Apoptosis 

Different cell death modes (the apoptotic rate) were evaluated in the HePG-2 after being treated with Ap or Ap-CH-BSA-FANPs quantitatively by the Annexin V/PI staining assay. Apoptotic cell death has considered a common destiny for all normal cells, but this route is inhibited in the malignancy cells. In apoptotic cell death mode, the cell is diminished after nucleus fragmentation, and then a distorted plasma membrane appears, which activates the phagocytosis process [55,56]. Although, in necrotic cell death mode, the cell was stretched because of the impermeability of the plasma membrane, then blown up, causing inflammation through all the nearby cells [57]. 

As illustrated in Figure 9B,C, the percentage of apoptotic death mode (early and late apoptosis) in the HePG-2 cells tresated with the IC_50_s of the Ap or Ap-CH-BSA-FANPs were 14.41% and 37.81%, respectively. So, the apoptotic percentage of HePG-2 cells treated with Ap-CH-BSA-FANPs was significantly increased than its percentage in Ap treatment (*p* = 0.001). These results suggested the high ability of chitosan to deliver the Ap into the cancer cells compared to the free Ap (Figure 9D). No significant change was observed in the necrotic phase between the Ap and the Ap-CH-BSA-FANPs. These data proved the mode of action of encapsulated apigenin was mostly via apoptosis, not necrosis, and when loading the Ap on the chitosan and coating it with BSA conjugated with folic acid, it increased the anticancer activity via enhancing its apoptotic action. Our results also agreed with recent studies revealing that loading of apigenin on liposome and PLGA could enhance its anticancer effect via inducing apoptosis [20,58]. In order to confirm that mode of action, the cell cycle was determined on the HePG-2 cells treated with Ap or Ap-CH-BSA-FANPs.

### 3.6. Ap-CH-BSA-FANPs Induced Cell Cycle Arrest 

Cell division is the frequent step that arises during replication with definite phases. These phases start with G1 (Gap 1), followed by the S (synthesis) phase, G2 (interphase) and finally, the M phase (mitosis). Thus, we analyzed the cell death mode to prove the way it is induced by Ap or Ap-CH-BSA-FANPs in HePG-2 cells. The HePG-2 was treated with the IC_50_s of Ap or Ap-CH-BSA-FANPs then the cell cycle was evaluated by flow-cytometer using PI. The calculated cells of every cell cycle phase were studied and expressed as % of the total cells. The increased S phase percentage was correlated with DNA duplication, and when arrested, it referred to the combination of the anticancer drug with DNA during synthesis or DNA fragmentation and hence apoptosis [59,60]. Similarly, the increased pre-G1 phase was correlated with the late apoptotic death of cancer cells [61]. The G2 phase is thought to act as a checkpoint for any mutations or damage in newly replicated DNA [62]. Increased G2 percentages were associated with increased DNA damage and the inability of cells to enter mitosis. An increased G2/M phase percentage means that the DNA damage is irreversible and cannot be repaired [63]. 

As shown from Figure 10D, the HePG-2 cells treated with Ap were arrested in the G2/M with a significant increase in the pre-G1 phase compared to both encapsulated Ap. and the untreated cells. That increase in the G2/M has inhibited the cells from entering another cell cycle, indicating more DNA damage (Figure 10B). Our findings were consistent with other in vitro studies that showed Ap-induced cell cycle arrest in the G2/M phase in various cancer cells [64,65,66,67,68]. However, the HePG-2 cells treated with Ap-CH-BSA-FANPs were arrested in the S phase as well as increased in the pre-G1 phase compared with the untreated cells (Figure 10C). Our data may suggest the way that Ap-CH-BSA-FANPs enter the nucleus and their combination with the DNA double strand, which prevents its synthesis and accumulates the cells in the S phase. On the other hand, Banerjee et al. (2017) claimed that apigenin-loaded liposomes triggered cell cycle arrest at the G2/M phase, which was similar to that of free Ap [58]. However, our data revealed that HePG-2 cells treated with encapsulated apigenin were correlated with apoptosis, and the encapsulation of Ap with BSA conjugated with folic acid totally changed its cell cycle action. In order to confirm the apoptotic pathway of the encapsulated apigenin on HePG-2 cells, we demonstrated the p53, MMP9, caspase 9, and Bcl-2 gene expressions. 

### 3.7. Ap-CH-BSA-FANPs Treatment Up-Regulated p53 and Caspase 9 and Down-Regulated MMP9 Genes Expression

The tumor suppressor gene (p53) is a protein that is stimulated by numerous types of cellular instabilities to protect the body cells from mutation. Cell-cycle arrest can then be initiated, resulting in the repair of damaged DNA and the induction of apoptotic cell death in mutated cells [69,70]. HCC can alter the p53 gene by blocking its activity, and hence that mutation can be analyzed as a prognostic marker of HCC [71,72,73]. Previous studies reported the increased expression level of p53 in cancer cells treated with pure Ap [74,75] and in liposomal Ap [58]. Our data showed significant p53 gene up-regulation (*p ≤* 0.0001, for both) in the HePG-2 cell treated with either Ap or Ap-CH-BSA-FANPs compared with the untreated cells. That increase illustrated the triggered activity of the p53 gene in the HePG-2 cells by the Ap or Ap-CH-BSA-FANPs treatments, with the greatest activity in the Ap-CH-BSA-FANPs treatment compared with the Ap treatment (*p ≤* 0.0001). That p53 activity accompanied by Ap-CH-BSA-FANPs treatment can induce cell-cycle arrest in the S phase and hence apoptosis, which explains our former results of arresting the cell cycle and enhancing apoptosis. 

MMPs (matrix metalloproteinases) control the dissociation and rearrangement of extracellular matrix constituents like collagen. The most important MMP is the MMP9. MMP9 plays an important role in the proteolytic dissociation of the extracellular matrix, degrading membrane peptides and denaturing cell extracellular proteins. That protein denaturation encourages cancer cell proliferation and hence metastasis [76]. MMP9 was found in a variety of cancers at very high levels [77,78,79,80]. Previous studies on Ap established its anti-metastatic effects via down-regulation of MMP-9 expression level [64,81,82,83,84]. Ganguly et al. (2021) reported that the apigenin-loaded PLGA can down-regulate the expression level of MMP-9 [20]. Our data showed low expression in the MMP9 gene of the HePG-2 cell treated with Ap-CH-BSA-FANPs compared with the untreated cells (*p =* 0.003) (Figure 11). The reduction in MMP9 has confirmed the Ap-CH-BSA-FANPs’ anti-proliferative effect against HePG-2 cells, and thus it may prevent cancer metastasis in vivo. However, no significant change in MMP9 gene expression was observed in the HePG-2 cells treated with Ap compared with the untreated cells (*p* = 0.011), which confirmed that the encapsulated apigenin could act as an anti-proliferative agent through an alternative pathway than the Ap pathway. 

Caspases, or cysteinyl aspartate-specific proteases, are enzymes that have been stimulated before apoptotic death to degrade the DNA and the membrane of the injured cells. Caspase gene stimulation has been considered the essential molecular incident that recognizes apoptotic cell death. The cell had two central apoptotic signaling pathways: extrinsic and intrinsic, also known as the mitochondrial pathway [85]. Both pathways have triggered caspase 8, 9, or 10 stimulation, which promotes the caspase 3 cleavage. Caspase 3 cleavage has the ability to fragment the DNA and denature the membrane proteins, leading to apoptosis [86]. The mitochondrial pathway is accompanied by cytochrome c activation and ends with caspase 9 stimulation [87]. Previous studies have shown that treating cancer cells with Ap increases caspase 9 expression [88,89].

Figure 12 showed significant overexpression of the caspase 9 gene (*p* = 0.002 and *p ≤* 0.0001, respectively) in the HePG-2 cell treated with either Ap or Ap-CH-BSA-FANPs compared with the untreated cells. The most significant overexpression of the caspase 9 gene (*p* = 0.001) was observed in the HePG-2 cell treated with Ap-CH-BSA-FANPs compared with Ap-treated cells. These results confirmed that Ap-CH-BSA-FANPs stimulated the p53 gene, ending with caspase 9 overexpression, leading to HePG-2 apoptotic cell death. Moreover, Ap-CH-BSA-FANPs were more potent than Ap in HePG-2 cytotoxicity and induction of apoptosis through activation of the p53 pathway. 

### 3.8. Ap-CH-BSA-FANPs Treatment Down-Regulated Bcl-2 in a Concentration Dependent Manner

B-cell lymphoma 2 (*Bcl-2*) is an anti-apoptotic protein that plays a central role in preventing apoptosis. *Bcl-2* inhibition has been linked to the development of effective anticancer drugs [90]. *Bcl-2* plays a critical role in both the intrinsic and p53 pathways. P53 has the ability to down-regulate the *Bcl-2* protein, thereby inhibiting cancer cell proliferation and increasing apoptosis [91]. In our study, we intended to prove that the apoptotic pathway of the Ap-CH-BSA-FANPs is through p53/caspase 9 pathway. So, we analyze the *Bcl-2* level in HePG-2 cells treated with 1/8, 1/4, 1/2, and IC_50_ values of either Ap or Ap-CH-BSA-FANPs. Our data from the IHC examination of the HePG-2 cells treated with encapsulated apigenin showed decreased expression of *Bcl-2* in a concentration-dependent manner (Figure 12) which confirmed the apoptotic pathway through p53/*Bcl-2*/caspase 9 pathway. Moreover, the HePG-2 cells treated with Ap showed decreased expression of *Bcl-2* in a concentration dependent manner (Figure 11F–I), confirming the same apoptotic pathway. Our findings agreed with a recent study that found that cancer cells treated with pure Ap or Ap-loaded PLGA had decreased *Bcl-2* protein expression [20]. 

### 3.9. Ap-CH-BSA-FANPs Treatment Increased SOD Level and Inhibited CAT Activity 

The metabolic activities of cancer cells were always higher than normal cells because of their high multiplication rate. That consequently elevated the production of reactive oxygen species (ROS). ROS play an essential role in cancer cell metabolism. Excess ROS production can help cancer progression by initiating DNA impairment and reprogramming cell metabolism [92]. Antioxidant enzymes were largely secreted from the cell to equilibrate the elevated ROS. The superoxide dismutase (SOD) enzyme produced within healthy cells can convert superoxide radicals to H_2_O_2_. Excess induction of SOD in cancer cells can induce apoptosis [93]. In contrast, increased catalase (CAT) can protect the tumor cell from excess ROS. By inhibiting CAT activity, the cancer cell’s death is increased via apoptosis [94]. 

Apigenin is a polyphenolic natural product that shows antioxidant activity by improving antioxidant enzyme levels like SOD and CAT [11]. Previous studies revealed that apigenin increases the SOD level and decreases the CAT level [95]. Moreover, the antioxidant properties of apigenin were increased after loading on albumin [96]. Both SOD and CAT activities were assessed to illustrate the influence of either Ap or Ap-CH-BSA-FANPs treatment on HePG-2 cells. As shown in Figure 13, SOD activity was increased by 2.4 times in HePG-2 cells treated with Ap-CH-BSA-FANPs compared with the untreated cells. On the other hand, CAT activity was inhibited by 59.36% in HePG-2 cells treated with Ap-CH-BSA-FANPs compared with the untreated cells. Furthermore, SOD activity was increased by 1.84 times and CAT activity was inhibited by 27.13% in HePG-2 cells treated with Ap compared with the untreated cells. These results demonstrated the higher antioxidant activity of the Ap-CH-BSA-FANPs than the Ap itself. Moreover, the anticancer effect of the encapsulated apigenin may be due to its ability to increase SOD and inhibit CAT, which enhances the apoptotic death of the HePG-2 cells.

## 4. Discussion 

Hepatocellular carcinoma is the world’s sixth most common cancer and the third most common cause of cancer-related mortality. Radiation and chemotherapy are the most popular treatments for HCC. The most popular chemotherapeutic drugs used to treat HCC are doxorubicin and cisplatin [3,4]. Nephrotoxicity and cardiotoxicity were the major side effects of the chemotherapeutic drugs that made it hard to treat HCC effectively [5,6]. Chemotherapy has been proven to be cytotoxic to vital organs as well as to cancer cells. Nanotechnology has recently become a popular method for drug delivery. These delivery methods have several advantages, including enhanced solubility of insoluble drugs and hence enhance their penetration into cancer cells. Targeting cancer cells with nano-materials uses strategies that disrupt the regulation of cancer cell proliferation and invasion with no effect on healthy cells. 

Apigenin is a popular flavonoid with anticancer and antioxidant properties [10,11,12,13]. Apigenin delivery was recently carried out by encapsulation with ChNPs-stearate nanogel, liposomes, or PLGA nanoparticles [19,20,33]. Because of the unique and specific features of hydrogels, nanogels are one of the controlled drug delivery technologies, especially for delivering bioactive compounds. Ap was coated with ChNPs-stearate, and its higher cytotoxicity and apoptotic activity were proved in vitro [33]. However, the low solubility of chitosan at physiological pH continues to be a concern in drug diffusion in vivo. Furthermore, there are a lot of limitations to using liposomes or PLGA nanoparticles for drug delivery, including high cost, oxidation of the loaded molecule, and insufficient drug loading [23,24,25]. 

Several studies established the enhanced cytotoxic activity of Ap against cancer cells after being loaded on different nano-carrier systems [21,25,49,58,97]. In our study, we loaded the Ap onto ChNPs, then coated them with albumin-folic to produce water-soluble nanoparticles and targeted the cancer cells by increasing the penetration of Ap inside the HePG-2 cell. The resultant encapsulated apigenin (Ap-CH-BSA-FANPs) agreed with other studies in showing a higher cytotoxic effect on both HePG-2 and Hela cells compared to the free Ap. A synergistic effect was confirmed when apigenin was combined with DOX, reducing DOX toxicity with increased efficiency [9]. In our study, the combination of Ap-CH-BSA-FANPs with DOX has a highly cytotoxic effect on HePG-2 cells, much like the cytotoxic effect of Cis. These data confirmed that encapsulated apigenin may be used together with chemotherapeutic drugs or could be a substitution for them.

Anticancer drugs kill cancer cells either via apoptosis or necrosis. Apoptosis is favorable because it happens without inflammation and can enhance phagocytosis. The Annexin V/PI stain can be used to detect apoptotic or necrotic stages. In the previous literature, Ap-loaded PLGA, nanogel, and liposomes had greater apoptotic effects than free Ap [20,21,97]. Similarly, in our study, the Ap-CH-BSA-FANPs had increased apoptotic action compared to pure Ap. 

Arresting the cell cycle at the G2/M phase was carried out by both the apigenin-loaded liposomes and free Ap [58]. Our current findings confirmed that Ap-CH-BSA-FANPs arrest the cell cycle at the S phase, demonstrating the penetration of our compound into the nucleus and its interaction with the DNA, preventing synthesis.

p53 is a tumor suppressor gene that protects against cellular mutation [69,70]. HCC can block p53 activity [98,99,100]. Cell-cycle arrest can be enabled by p53. p53 can also fix the damaged DNA, causing apoptotic cell death in mutant cells. Previous research has found that cancer cells treated with free Ap or Ap-loaded PLGA have higher levels of p53 expression [20,74,75]. In the present study, both free Ap and Ap-CH-BSA-FANPs had elevated p53 gene expression, with higher levels in the Ap-CH-BSA-FANPs. Ganguly et al. (2021) established the down-regulation of MMP-9 expression in rat liver treated with apigenin-loaded PLGA [20]. Similarly, our data revealed MMP9 down-regulation in the HePG-2 cell treated with Ap-CH-BSA-FANPs, but no change in MMP9 levels in HePG-2 treated with free Ap. This data shows that the Ap-CH-BSA-FANPs may be much more effective than Ap in preventing cancer metastasis in vivo.

Apoptotic cell death is always complemented by caspase activation, particularly caspase 9 (which triggers caspase 3 cleavage) and, finally, DNA fragmentation [86]. Ap treatment was accompanied by increased caspase 9 expression in cancer cells [88,89]. Our present study revealed caspase 9 overexpression in the HePG-2 cells treated with Ap-CH-BSA-FANPs compared with those treated with Ap. These findings revealed that Ap-CH-BSA-FANPs were also more effective than Ap in inducing apoptosis in HePG-2 cells via activating the p53 pathway, resulting in caspase 9 up-regulation.

Ganguly et al. (2021) proved that treatment with either free Ap or Ap-loaded PLGA decreased Bcl-2 protein expression in cancer cells [20]. Similarly, our findings indicated decreased expression of *Bcl-2*, in a concentration-dependent manner, in HePG-2 cells treated with either Ap or Ap-CH-BSA-FANPs, which established the p53/*Bcl-2*/caspase 9 apoptotic pathway. In addition, the antioxidant potential of Ap-loaded nano-materials was further increased than the free Ap [95,96,97]. In our study, Ap-CH-BSA-FANPs were shown to increase SOD activity with inhibited CAT activity in HePG-2 cells more than in Ap treatment. This suggests that its potential as an anticancer agent is due to increased SOD levels and inhibition of CAT, leading to enhanced apoptotic death.

## 5. Conclusions

For the first time, the Ap was loaded onto chitosan nanoparticles and then coated with folic acid conjugated BSA to help with targeted delivery and enhanced solubility in physiological pH compared with both Ap-loaded PLGA and liposomes. The Ap-CH-BSA-FANPs were produced by a simple method with low-cost materials that could be available for increased production. Moreover, our study demonstrated the anticancer effect of the Ap-CH-BSA-FANPs against HepG-2 and Hela cell lines with synergetic potency when combined with DOX. Furthermore, the apoptotic effect of the Ap-CH-BSA-FANPs on the HepG-2 cells was demonstrated by stimulating *p53* expression and arresting the cell cycle at the S phase. The stimulated *p53* gene will end with caspase 9 overexpression, leading to HePG-2 apoptotic cell death. The reduced *Bcl-2* expression of the HepG-2 cells treated with Ap-CH-BSA-FANPs confirmed the p53/*Bcl-2*/caspase 9 apoptotic pathway. Moreover, the effect of Ap-CH-BSA-FANPs on cancer metastasis was revealed by the decrease in MMP9 expression. Our data also confirmed the anti-proliferative effect of the Ap-CH-BSA-FANPs through increasing SOD levels and decreasing the CAT concentration, which enhances the apoptotic death of the HePG-2 cells. Therefore, Ap-CH-BSA-FANPs were easy to produce with low coast, continued drug release, high solubility in physiological pH, good loading capacity, and were more stable than the formerly Ap-loaded liposomes or PLGA. Ap-CH-BSA-FANPs, as a natural nano-compound, might be a promising anticancer agent for the treatment of hepatocellular carcinoma. However, in vivo experiments must be done to confirm the safety of the Ap-CH-BSA-FANPs. 

## Data Availability

Data available in a publicly accessible repository.

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
