# Peer review of "The Effect of Encapsulated Apigenin Nanoparticles on HePG-2 Cells through Regulation of P53"

_pharmaceutics, 2022, doi:10.3390/pharmaceutics14061160_

Round 1
Reviewer 1 Report
In this work, Zayed et al. produced Chitosan Nanoparticles loaded with Ap and coated with FA-BSA for application in cancer therapy. Before publication, authors must address the following issues:
The release studies must be performed in PBS pH 7.4 (physiologic) and 6.5 (tumor microenvironment)
The novelty of this work must be highlighted.
No data confirming the successful synthesis of FA-BSA was provided.
The full DLS data must be provided (xx axis from 1 to 10 000 nm).
“This section may be divided by subheadings. It should provide a concise and precise description of the experimental results, their interpretation, as well as the experimental conclusions that can be drawn.” This sentence is from the Pharmaceutics Template?
The release studies must be reported as Cumulative release (%). Moreover, longer time points are required (at least 48h).
The biocompatibility of the blank nanoparticles (without Ap) must be reported.
The discussion must be improved. Authors must justify their findings with literature reports.
The targeting capacity of the nanoparticles was not demonstrated.
Author Response
Reviewer 1
Comments and Suggestions for Authors
In this work, Zayed et al. produced Chitosan Nanoparticles loaded with Ap and coated with FA-BSA for application in cancer therapy. Before publication, authors must address the following issues:
Question 1: The release studies must be performed in PBS pH 7.4 (physiologic) and 6.5 (tumor microenvironment)
Response to Question 1: We thank the reviewer 1 so much for his/her great comment, and we strongly agree with the reviewer 1 that cumulative drug release in physiological and tumor pH would bring relevant information on the % of drug release of the developed systems.
The in vitro release study of the Ap from Ap-CH-BSA-FANPs and pure apigenin was performed according to the method of Hanafy et al. 2021 [37,38]. The Ap pure and encapsulated Ap were released in PBS (pH 7.4) and PBS (pH 6.5) containing 1% Tween 80 for more than 50 h. Briefly, 16 mg of Ap pure and (Ap-CH-BSA-FANPs) suspensions were transferred into dialysis bags (molecular weight 12,000–14,000Da) and then immersed in 40 mL release medium at 37°C with continuous stirring at 100 rpm. Following that, 1 mL was aspirated from each baker to determine the quantity of apigenin released at 0, 1, 2, 4, 6,12,24 and 48 hours. At the same time, the baker's solution was replaced after every aspiration with the same volume of freshly prepared release medium. Its absorbance was measured using a UV-Vis spectrophotometer at a wavelength of 356 nm.
The % of cumulative drug release was measured using the constructed standard calibration curve. Mean release of three measurements was used to evaluate the drug release for each of point.
Question 2: The novelty of this work must be highlighted.
Response to Question 3: first We would like to thank the reviewer 1 so much for his/ her comment.
The novelty of the current work was highlighted
Question 3: No data confirming the successful synthesis of FA-BSA was provided.
Response to Question 3: We thank the reviewer 1 so much for this great comment, which give us the possibility to improve the manuscript.
Successful attachment of FA and BSA was studied previously by Hanafy, et al.
- Hanafy, N.A.N., Quarta, A., Di Corato, R. et al.Hybrid polymeric-protein nano-carriers (HPPNC) for targeted delivery of TGFβ inhibitors to hepatocellular carcinoma cells. J Mater Sci: Mater Med 28, 120 (2017). https://doi.org/10.1007/s10856-017-5930-7
- Hanafy NAN, Leporatti S, El-Kemary M. Mucoadhesive curcumin crosslinked carboxy methyl cellulose might increase inhibitory efficiency for liver cancer treatment. Mater Sci Eng C Mater Biol Appl. 2020 Nov;116:111119. doi: 10.1016/j.msec.2020.111119.
- Hanafy NAN, Leporatti S, El-Kemary MA. Extraction of chlorophyll and carotenoids loaded into chitosan as potential targeted therapy and bio imaging agents for breast carcinoma. Int J Biol Macromol. 2021 Jul 1;182:1150-1160. doi: 10.1016/j.ijbiomac.2021.03.189.
However, new FTIR experiment was added to the result in Fig.5;B. the spectrum of BSA-FA showed band at 1702 cm-1 that was attributed to carboxylic group of FA. Additionally, the characteristic bands of p aminobenzoic acid were observed at 1404 cm-1 to 1100 cm-1
Question4: The full DLS data must be provided (xx axis from 1 to 10 000 nm).
Response to Question 4: We thank the reviewer1 so much for this great and valuable comment. Following the reviewer´s suggestion, we provide our sorry that the data obtained by our instrument is mostly controlled by Brookhaven program. Since, NPs were determined using dynamic light scattering (DLS) technique by a Brookhaven instruments 90Plus particle size. The software uses Stokes–Einstein equation to transform diffusion coefficients, determined by dynamic light scattering, into hydrodynamic diameters presented as measurement results according to (Nazarenko et al.,2011). For this reason, the axis of scale bar is controlled mechanically according to diameter of sample during its measurement. Meanwhile, the distribution of NPs in solution was provided. While, Malvern instrument may provide axis from 1 to 10 000 nm and this instrument is not located in our institute.
Nazarenko Y, Han TW, Lioy PJ, Mainelis G. Potential for exposure to engineered nanoparticles from nanotechnology-based consumer spray products. J Expo Sci Environ Epidemiol. 2011;21(5):515-528. doi:10.1038/jes.2011.10
Question 5: “This section may be divided by subheadings. It should provide a concise and precise description of the experimental results, their interpretation, as well as the experimental conclusions that can be drawn.” This sentence is from the Pharmaceutics Template?
Response to Question 5: We thank the reviewer 1 for this great comment. This sentences were written in section of results in Pharmaceutics Template, line 53. According to the structure of journal template, the results were covered all obtained experiments and were divided into subheadings. Additionally, discussion was further being separated.
Question 6: The release studies must be reported as Cumulative release (%). Moreover, longer time points are required (at least 48h).
Response to Question 6: We thank the reviewer 1 so much for this great comment, which give us the possibility to improve the manuscript.
The % drug release was calculated and a graph of % drug release against time was plotted, release studies were performed in triplicate for each formulation. Data were expressed as the cumulative amount of apigenin permeated through the membrane considering the total amount of drug applied of each formulation. The result was written as following;
An in vitro study revealed that encapsulated apigenin exhibited continuous release during incubation at time intervals. This strategy depends mainly on the amount of apigenin entrapped in chitosan moieties and the interaction of the amino groups of chitosan and the hydroxyl groups of apigenin after their de-protonation. Due to its hydrophobic nature, apigenin pure showed good release in presence of chemical imitator such as DMSO. The drug release was 40 % after 24 h incubation. While, less amount of release was obtained by Ap-CH-BSA-FANPs (20%) in presence of the same initiator. This is due to the chemical interaction between entrapped Ap inside moieties of chitosan and BSA. Contritely, slow release was observed by free apigenin at physiological pH 7.4 and tumor microenvironment pH 6.5. This is due to its solubility in aqueous solution. While, the release of Ap from nano-formulation (Ap-CH-BSA-FANPs) was quite higher in buffer of pH 6.5 compared to PBS (pH 7.4).
% cumulative drug release of encapsulated Ap. was around 85% at pH 6.5 while that was around 60% (at pH 7.4) at the end of 50h incubation. The hydroxyl groups of apigenin interacted with the amino group of the chitosan forming stable structure. such this interaction can be ionized at acidic pH, thereby releasing the drug effectively at acidic pH suitable to target cancer cells
In the current study, apigenin was released continuously throughout the time of incubation, providing control release compared to free apigenin (Fig. 6). In the meantime, our results revealed the possible release of apigenin encapsulated inside moieties of chitosan after its cellular uptake.
Arcot RC, Chan J, Choong T ,Teeba M , Selvadurai M, Sokkalingam A. 2011. In-vitro studies and evaluation of Metformin marketed tablets-Malaysia. Journal of Applied Pharmaceutical Science, 1:214-217.
Concentration of drug (µg/ml)= (slope × absorbance) ± intercept
Amount of drug = Concentration × Dissolution bath volume × dilution factor released mg/ ml 1000
Cumulative percentage = Volume of sample withdrawn (ml) × P (t – 1) + Pt release (%) Bath volume
(v) Where Pt = Percentage release at time t Where P (t – 1) = Percentage release previous to‘t’
Question 7:The biocompatibility of the blank nanoparticles (without Ap) must be reported.
Response to question 7: We would like to thank reviewer1 so much for his/her great comment.
The blank nanoparticles without AP was investigated and the result was added to fig. 8 and table 1. It was written as Free capsules
Question 8: The discussion must be improved. Authors must justify their findings with literature reports.
Response to question 8: Discussion was being separated from section of (result and discussion). The current findings were described and discussed according to previous literature.
Question 9: The targeting capacity of the nanoparticles was not demonstrated.
Response to Question 9: We would like exactly to thank reviewer1 for his/her great comment. The targeting capacity of Ap-CH-BSA-FANPs internalized inside HePG2 cells was measured by fluorescence microscopy. Since, R6G labelled Ap-CH-BSA-FANPs (Fig.7; A &B) were successfully localized inside the cytoplasm as demonstrated by the intensity of fluorescence emission of red colour located in the perinuclear region compared to Ap-CH-BSA-NPs without FA (Fig.7;C)
Fluorescence images clearly demonstrate that Ap-CH-BSA-NPs conjugated to FA were readily accumulated in cellular compartments leading to increase drug capacity and efficiency.

Reviewer 2 Report
The present manuscript entitled "The effect of encapsulated apigenin nanoparticles on HePG-2 cells through regulation of P53" shows anticancer activity of apigenin against liver cancer cells. The authors showed that the anticancer activity of apigenin was further increased when the drug was incorporated in chitosan-bearing nanoparticles. The study was performed meticulously and extensively conducted. The authors should respond to the following comments:
- There are several grammatical errors and many sentences should be edited. For example, lines 108-110 and lines 138-140
- 2.3. Fabrication of apigenin-NPs: line 179, is it 40 ml of chitosan or 4 ml of chitosan solution?
- line 207, it should be percent encapsulation efficiency
- line 203, 209. why two different wavelengths were used used to estimate the quantity of apigenin.
- Results section: lines 368-369, The authors state that the negative zeta potential of favors their RES uptake by liver and spleen. How it it beneficial for the treatment. Once nanoparticles are taken up by RES , they may be cleared from the system.
- Drug release: The authors showed the drug release till 6 hours. It should be extended beyond 6 hours, up to 24 hours.
- An important control is missing. Why did authors not use Sham chitosan nanoparticles in this study? Moreover, it would have been interesting to analyze the effect of DOX or Cis-loaded chitosan nanoparticles.
Author Response
Reviewer 2
Comments and Suggestions for Authors
The present manuscript entitled "The effect of encapsulated apigenin nanoparticles on HePG-2 cells through regulation of P53" shows anticancer activity of apigenin against liver cancer cells. The authors showed that the anticancer activity of apigenin was further increased when the drug was incorporated in chitosan-bearing nanoparticles. The study was performed meticulously and extensively conducted. The authors should respond to the following comments:
First: We would like to thank reviewer 2 for his /her great and valuable comments.
Question 1: There are several grammatical errors and many sentences should be edited. For example, lines 108-110 and lines 138-140
Response to question 1: We would like to thank reviewer2 for his /her great comment. The sentences were corrected.
Question 2: 2.3. Fabrication of apigenin-NPs: line 179, is it 40 ml of chitosan or 4 ml of chitosan solution?
Response to question 2: We would like to thank reviewer 2 for his/ her great comment. The fabrication of apigenin NPs were corrected as following;
After that, 3 mg of apigenin was dissolved in 10 mL of 1N NaOH and dropped slowly into 10 mg/40 mL of chitosan under continuous stirring for 30 min.
Question 3: line 207, it should be percent encapsulation efficiency
Response to question 3: We would like to thank reviewer2 for his /her valuable comment. Encapsulation efficiency (%) was added
Question 4:line 203, 209. why two different wavelengths were used to estimate the quantity of apigenin.
Response to question 4: We would like to thank reviewer2 for his /her great comment. Apigenin considers from flavones that can exhibit two major absorption bands: Band I (320–385 nm) represents the B ring absorption, while Band II (250–285 nm) corresponds to the A ring absorption. Functional groups attached to the flavonoid skeleton may cause a shift in absorption. In the current study, amount of apigenin was estimated at 270 nm in all experiments.
Kumar S, Pandey AK. Chemistry and biological activities of flavonoids: an overview. ScientificWorldJournal. 2013 Dec 29;2013:162750. doi: 10.1155/2013/162750. PMID: 24470791; PMCID: PMC3891543.
Question 5: Results section: lines 368-369, The authors state that the negative zeta potential of favors their RES uptake by liver and spleen. How it it beneficial for the treatment. Once nanoparticles are taken up by RES , they may be cleared from the system.
Response to question 5: We would like to thank reviewer2 for his/her great comment. It is reported that slightly negative charge may be introduced to the NPs surface to reduce the undesirable clearance by the reticuloendothelial system
Xiao K, Li Y, Luo J, Lee JS, Xiao W, Gonik AM, Agarwal RG, Lam KS. The effect of surface charge on in vivo biodistribution of PEG-oligocholic acid based micellar nanoparticles. Biomaterials. 2011 May;32(13):3435-46. doi: 10.1016/j.biomaterials.2011.01.021.
Question 7: Drug release: The authors showed the drug release till 6 hours. It should be extended beyond 6 hours, up to 24 hours.
Response to question7: The drug release (%) was extended into 24 h. Additionally, drug release (%) in physiological pH 7.4 and in tumour microenvironment pH 6.5 was investigated until incubation for 48 h.
Question 8: An important control is missing. Why did authors not use Sham chitosan nanoparticles in this study? Moreover, it would have been interesting to analyze the effect of DOX or Cis-loaded chitosan nanoparticles.
Response to question 8: In the current study, Free capsules (without apigenin) was used in most of experiments to provide clear comparison between free capsules and encapsulated one. Free capsules were used in FTIR, UV Visible spectrophotometer, Xray, and MTT assay for cytotoxicity experiment. Meanwhile, we provide our sorry, because this manuscript, is subjected to MSc research. The effect of DOX or Cis-loaded chitosan nanoparticle cannot be studied at moment. This great idea will be planned in our future work.

Round 2
Reviewer 1 Report
The caption of Figure 7 is incomplete.
Author Response
Comments and Suggestions for Authors
First, we would like to thank reviewer 1 for his/her great comment.
Question 1: The caption of Figure 7 is incomplete.
Response to question 1:the cation of Figure 7 was improved
Fig.7: UV visible of R6G conjugated to Ap-CH-BSA-NPs or Ap-CH-BSA-FANPs (A). Fluorescence image of Ap-CH-BSA-FANPs conjugated R6G (B). Qualitative cell internalization assay using R6G conjugated different nanoparticles by using fluorescence microscopy (C), Fluorescence images demonstrate cellular internalization of Ap-CH-BSA-FANPs conjugated R6G in HePG-2 cells; a) crystal violet to show cell morphology. b) grayscale image of crystal violet. c) TRIC channel of R6G d) merge between grayscale image and TRIC channel by using Image j program. Fluorescence images demonstrate cellular internalization of Ap-CH-BSA-NPs conjugated R6G in HePG-2 cells a) crystal violet to show cell morphology. b) grayscale image of crystal violet. c) TRIC channel of R6G d) merge between grayscale image and TRIC channel by using Image j program.

Reviewer 2 Report
The authors have responded to all the comments.
Author Response
Comments and Suggestions for Authors
The authors have responded to all the comments.
We would like to thank reviewer 2 for his /her great time spending to revise our manuscript.
Thank you so much.